# SmartCache: Context-aware Semantic Cache for Efficient Multi-turn LLM Inference

**Chengye Yu**
Chinese University of Hong Kong
Hong Kong, China
cyyu23@cse.cuhk.edu.hk

**Tianyu Wang** ✉
Shenzhen University
Shenzhen, China
tywang@szu.edu.cn

**Zili Shao**
Chinese University of Hong Kong
Hong Kong, China
shao@cse.cuhk.edu.hk

**Song Jiang**
University of Texas at Arlington
Arlington, USA
song.jiang@uta.edu

## Abstract

Large Language Models (LLMs) for multi-turn conversations suffer from inefficiency: semantically similar queries across different user sessions trigger redundant computation and duplicate memory-intensive Key-Value (KV) caches. Existing optimizations such as prefix caching overlook semantic similarities, while typical semantic caches either ignore conversational context or are not integrated with low-level KV cache management. We propose SmartCache, a system-algorithm co-design framework that tackles this inefficiency by exploiting semantic query similarity across sessions. SmartCache leverages a Semantic Forest structure to hierarchically index conversational turns, enabling efficient retrieval and reuse of responses only when both the semantic query and conversational context match. To maintain accuracy during topic shifts, it leverages internal LLM attention scores—computed during standard prefill—to dynamically detect context changes with minimal computational overhead. Importantly, this semantic understanding is co-designed alongside the memory system: a novel two-level mapping enables transparent cross-session KV cache sharing for semantically equivalent states, complemented by a semantics-aware eviction policy that significantly improves memory utilization. This holistic approach significantly reduces redundant computations and optimizes GPU memory utilization. The evaluation demonstrates SmartCache's effectiveness across multiple benchmarks. On the CoQA and SQuAD datasets, SmartCache reduces KV cache memory usage by up to $59.1\%$ compared to prefix caching and $56.0\%$ over semantic caching, while cutting Time-to-First-Token (TTFT) by $78.0\%$ and $71.7\%$, respectively. It improves answer quality metrics, achieving $39.9\%$ higher F1 and $39.1\%$ higher ROUGE-L for Qwen-2.5-1.5B on CoQA. The Semantic-aware Tiered Eviction Policy (STEP) outperforms LRU/LFU by $29.9\%$ in reuse distance under skewed workloads.

## 1 Introduction

Transformer-based Large Language Models (LLMs) [39], such as GPT [30], PaLM [8], and Llama [37, 16], are revolutionizing the world with their powerful and versatile capabilities, enabling a wide range of tasks ranging from question answering to code generation [45, 30, 47]. A particularly prominent application is multi-turn conversation [12, 20, 46, 18, 25], underpinning interactive systems where users engage with models over extended dialogues, often exploring specific topics in depth [36]. This

39th Conference on Neural Information Processing Systems (NeurIPS 2025).

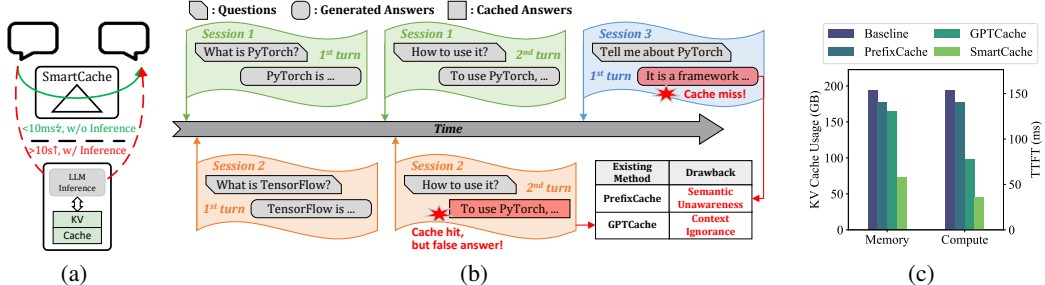

Figure 1: 1(a) illustrates the functionality of SmartCache. 1(b) shows an example of multi-user multi-turn conversations. GPTCache [6] would encounter a false positive cache hit for Session 2 Turn 2 due to flat query embedding search. PrefixCache [40] fails to reuse the answer of Session 1 Turn 1 for the semantically similar query of Session 3 Turn 1 due to exact token match. 1(c) shows the resource consumption of different methods on SQuAD dataset with LLama-3.1-8B model.

paradigm is central to applications like collaborative content analysis and comprehension, where users interact with materials like news articles or videos through conversational interfaces.

However, current LLM serving systems suffer from a fundamental inefficiency in this setting: semantic redundancy. Different users in separate sessions often ask questions that are semantically equivalent or highly similar. Existing systems treat each session independently. Semantically similar queries often trigger separate, full LLM inference passes, redundantly producing identical or nearly identical responses. This results in unnecessary GPU computation. Furthermore, the associated Key-Value (KV) caches—essential for efficient token generation yet highly memory-intensive—are maintained separately for each session, resulting in excessive GPU memory usage and added overhead from memory management tasks such as offloading [14, 19].

Existing optimization techniques are inadequate to address the problem. Prefix caching methods, proposed in RadixAttention [48] and commonly used in systems like vLLM [40, 19, 27, 15, 4, 1], leverage shared token prefixes (e.g., system prompts) to share initial KV cache blocks. They rely on exact token matching and fail to recognize semantic equivalence when queries are phrased differently (e.g., "What is PyTorch?" vs. "Tell me about PyTorch" as shown in Figure 1(b)). Their benefits are also largely confined to the initial prompt processing (prefill stage), offering little help for the time-consuming token-by-token decoding phase. On the other hand, higher-level semantic caching systems [6] operate by storing text responses indexed by query embeddings. While they capture semantic similarity, they often struggle in multi-turn scenarios due to two key limitations. (1) **Context Ignorance**. They often retrieve cached responses based solely on the current query's embedding, disregarding the preceding conversation turns. This leads to incorrect or nonsensical answers when a query's meaning is context-dependent (e.g., "How to use it?"). (2) **Decoupling from Inference State**. These caches operate above the LLM inference engine. A cache hit provides text but does not update the LLM's internal KV cache state. A subsequent cache miss can force the LLM to reprocess the entire conversation history to regenerate the KV cache, resulting in significant computational overhead and undermining potential memory savings. There is a clear need for a solution that understands semantics, preserves conversational context, and is tightly integrated with the underlying inference and memory management mechanisms. As depicted in Figure 1(c), utilizing semantics to further reduce resource usage remains largely unexplored.

To bridge this gap, we present SmartCache — a system-algorithm co-design framework purpose-built to optimize multi-turn LLM inference by leveraging cross-session semantic similarity while preserving contextual integrity. SmartCache tightly integrates a novel semantic caching layer with the core LLM serving engine and its KV cache management. It is built upon several key design principles and mechanisms.

• **Semantic Forest**. To ensure cached responses are used only in the appropriate context and to efficiently manage the branching nature of conversations, SmartCache employs the Semantic Forest data structure (Section 3.1). This hierarchy of trees organizes conversational turns (query-response pairs) semantically. Lookups for incoming queries are constrained to relevant branches based on the preceding turn, ensuring contextual accuracy and efficient search. Cache misses dynamically grow the forest, capturing new conversational paths.

- **Attention-based Context Identification**. Recognizing that conversations can dynamically shift topics, SmartCache incorporates an attention-based context identification module (Section 3.2). By analyzing attention patterns during standard query prefill—a low-overhead signal from the model's internals—SmartCache distinguishes between queries that extend the current context and those that begin a new one. This allows sessions to intelligently navigate or transition between different semantic trees, enhancing both cache effectiveness and response accuracy.

- **Semantic-aware Tiered Eviction/Prefetching**. To maximize resource savings, SmartCache features a co-designed KV cache sharing and eviction mechanism that deeply integrates the semantic layer with memory management (Section 3.3). A novel two-level mapping connects semantic nodes to physical KV cache blocks, allowing multiple sessions referencing the same conversational point to transparently share underlying GPU memory. Furthermore, instead of relying on blind page-level or session-level eviction, SmartCache employs lightweight and powerful **S**emantic-aware **T**iered **E**viction/**P**refetching (STEP) policy. STEP manages the lifecycle of KV cache blocks using factors derived from the Semantic Forest structure and usage patterns, including semantic depth, session affinity, and node popularity. This refined, contextually-informed approach optimizes memory utilization more effectively than traditional page- or session-based methods.

To evaluate the effectiveness of SmartCache, we conduct extensive experiments across multiple benchmarks and models. On the CoQA and SQuAD datasets, SmartCache reduces KV cache memory usage by up to $59.1\%$ compared to prefix caching and $56.0\%$ over semantic caching, while achieving $78.0\%$ and $71.7\%$ reductions in Time-to-First-Token (TTFT), respectively. SmartCache also improves answer quality metrics, attaining $39.9\%$ higher F1 and $39.1\%$ higher ROUGE-L scores for the Qwen-2.5-1.5B model on CoQA, demonstrating its ability to preserve contextual accuracy. Furthermore, our STEP policy outperforms traditional LRU/LFU policies by up to $29.9\%$ in reuse distance under skewed workloads, validating its efficiency in dynamic multi-session environments. These results highlight SmartCache's ability to balance computational efficiency, memory optimization, and answer quality across diverse conversational scenarios.

## 2 Background

### 2.1 Related Works

Several prior works have explored various methods to address the inefficiencies in multi-user multi-turn conversations. Table 1 compares different caching strategies used in multi-turn conversations.

Table 1: Comparison of Caching Strategies for Multi-Turn LLM Inference.

| | Baseline | PrefixCache | Flat Embedding | Context-aware Semantic Cache |
|---|---|---|---|---|
| Feature | No Reuse | e.g., vLLM APC [40], RadixAttention [48, 14] | e.g., GPTCache [6], SCALM [21] and [34] | **SmartCache** (ours) |
| Reuse Trigger | ✗ | Exact Prefix Match | Flat Similarity | **Hierarchical Semantic** |
| Context Aware | ✗ | ✗ | Limited† | ✓ (**Hierarchical**) |
| Resource Reused | ✗ | KV Cache* | Text Response only | **Text Response + KV Cache** |
| Computation Saving | ✗ | Limited | ✓ | ✓ |
| Integrated System/Algorithm | ✗ | ✗ | ✗ | ✓ (**Co-designed**) |
| Key Bottleneck | High Redundancy | No Semantic Reuse | Context Ignorance | - |
| Benefit Focus | Simplicity | Prefill Speedup | Saves Full Compute | Context Awareness + Memory/Compute Eff. |

†GPTCache can use history for embedding, but lacks explicit conversational structure.
*Only exactly matched prefix tokens are reused.

### 2.2 Problem Description

Let $\mathcal{S}$ be the set of active user sessions. Each session $s \in \mathcal{S}$ arriving at turn $k+1$ has a conversational history $H_s^{(k)}$ consisting of the preceding $k$ query-response pairs. We can represent this history as a sequence or concatenation: $H_s^{(k)} = \{(q_i, a_i)\}_{i=1}^{k}$ or, if treated as a single string for embedding purposes, $H_s^{(k)} \approx \|_{i=1}^{k} (q_i\|a_i)$, where $\|$ denotes string concatenation. When session $s$ issues the new query $q_{k+1}$, the goal is to efficiently determine the appropriate response $a_{k+1}$.

A standard LLM serving system computes $a_{k+1} = M(q_{k+1}|H_s^{(k)})$, where $M$ is the LLM and $H_s^{(k)}$ represents the relevant history context for session $s$. This computation is expensive.

A flat query-answer cache like GPTCache aims to find if there exists a previously asked query $q_{prev}$ (from any turn $j$ in any previous session) in a global set of all past queries $Q_{all}$ such that

$sim(E(q_{k+1}), E(q_{prev})) \geq \tau_{sim}$, where $E$ is an embedding function, $sim$ is a similarity metric, and $\tau_{sim}$ is a threshold. If such a $q_{prev}$ with associated response $a_{prev}$ is found, $a_{k+1}$ is set to $a_{prev}$.

$$\text{Find } (q_{prev}, a_{prev}) \text{ s.t. } q_{prev} \in Q_{all} \wedge sim(E(q_{k+1}), E(q_{prev})) \geq \tau_{sim}$$

The primary limitation is that it disregards the conversational context $H_s^{(k)}$. The semantic meaning of $q_{k+1}$ is often dependent on $H_s^{(k)}$, and reusing $a_{prev}$ generated under a different context $H_{s'}^{(k')}$ can be incorrect, even if $q_{k+1}$ and $q_{prev}$ are textually similar.

One might attempt to address this by directly incorporating the conversation history $H_s^{(k)}$ in the query embedding. A flat semantic cache could store embeddings of the form $e_{query\_hist} = E(H_s^{(k)}||q_{k+1})$. For an incoming query $q'_{k'+1}$ with history $H_{s'}^{(k')}$, the system would compute $e'_{query\_hist} = E(H_{s'}^{(k')}||q'_{k'+1})$ and search for a stored embedding $e_{stored} = E(H_{prev}^{(j)}||q_{prev})$ in a global index, such that $sim(e'_{query\_hist}, e_{stored}) \geq \tau_{sim}$.

While this approach includes history information, it suffers from the *Long History Dominance* [23, 32, 38, 22, 10] issue, especially as the number of turns $k$ increases. Standard embedding models struggle to effectively represent long concatenated sequences in a fixed-size vector such that the final query $q_{k+1}$ remains distinctly identifiable. As the history $H_{s'}^{(k')}$ grows long, the embedding vector $E(H_{s'}^{(k')}||q'_{k'+1})$ tends to be increasingly dominated by the content of the history $H_s$ itself, rather than the specific new query $q_{new}$, i.e. $E(H_{s'}^{(k')}||q'_{k'+1})$ is closer to $E(H_{s'}^{(k')})$ but more distant from $E(q'_{k'+1})$.

## 3 The Design of SmartCache

In this section, we introduce SmartCache, a context-aware semantic cache designed for multi-turn LLM inference with enhanced context-awareness. First, as illustrated in Figure 2, SmartCache organizes and indexes conversation contexts from different users into a Semantic Forest, as detailed in Section 3.1. The semantic forest comprises multiple individual semantic trees, where each conversation turn is represented as a semantic node. Semantic nodes can be associated with and reused across multiple conversation sessions, enabling efficient context sharing. Second, in Section 3.2, SmartCache introduces a semantic identification policy that leverages attention scores from the prefilling stage of user queries to identify non-

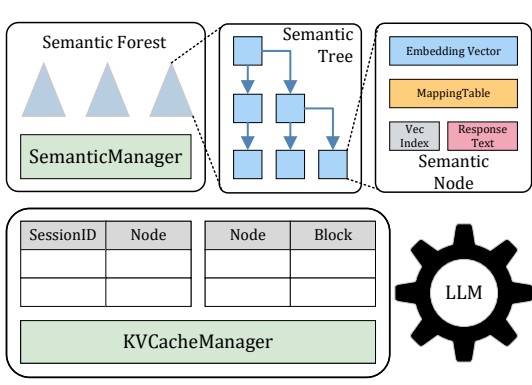

Figure 2: Overview of SmartCache.

contextual queries, to support dynamic context switching, and unlocks additional sharing opportunities. Finally, we present the transparent cross-session KV cache sharing design, coupled with a novel eviction policy, enabling efficient sharing of KV caches based on semantic similarity in Section 3.3.

### 3.1 Semantic Context Manager

#### 3.1.1 Semantic Tree

In a multi-turn conversational scenario, users often begin a session with a simple question on a specific topic, followed by a series of progressively deeper queries. Different users may explore the same topic by asking similar questions in varied ways or by focusing on different aspects. Such relationships can be effectively represented using a tree structure, referred to as a semantic tree in SmartCache. Each semantic tree organizes query-response turns as semantic nodes under a specific semantic context. The initial query of a new session becomes the root node of a new semantic tree, while subsequent queries create child nodes linked to the corresponding semantic parents.

Conversation sessions centered on the same topic can share a common semantic tree, allowing them to leverage cached responses when similar queries have been made by other users. Each conversation

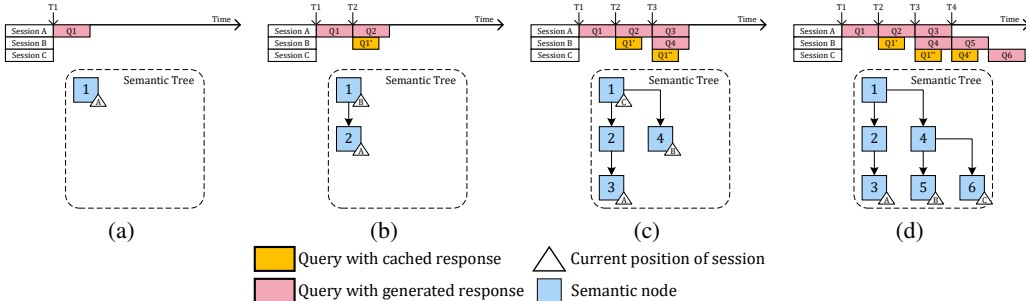

Figure 3: Example of a hierarchical semantic tree shared by three sessions (A, B, and C).

session is associated with a semantic node, representing the last query-response turn of the session. Users can ask progressive questions based on the context represented by the semantic node. In these cases, child nodes are evaluated to possibly reuse responses from similar queries, avoiding the need for regeneration. When there are no reusable responses, the query is processed by the LLM using the contextual history traced from the root to the current semantic node. In the case where a user poses an unrelated question and shifts to a different topic, SmartCache either locates a suitable existing semantic tree or initiates a new one to reflect the updated context.

Figur 3 illustrates a comprehensive example of a semantic tree shared across three sessions (A, B, and C). At time T1, session A begins with query Q1, which is processed by the LLM to generate a response, as it is the first query on the topic. Q1 and its generated response are cached as semantic node 1, which becomes the root of the newly created semantic tree. At T2, session B issues Q1′, a query semantically similar to Q1, and retrieves the cached response from semantic node 1. Meanwhile, session A raises query Q2 and receives a newly generated response. At T3, session C initiates Q1″ and retrieves the same cached response from semantic node 1. However, when session B raises query Q4, no similar query is found among the children of node 1, the current position of session B. As a result, its response is generated by the LLM, and a new branch from node 1 to node 4 is created. Session C continues to reuse node 4 until it encounters a cache miss at query Q6.

The semantic tree design benefits multi-turn conversations in two aspects. First, by reusing semantically similar nodes across different sessions, queries with identical or similar semantics are computed and decoded by the LLM only once, and their responses are shared. This approach significantly reduces computational overhead and alleviates memory pressure, as the KV cache of common semantic nodes is shared among sessions. Second, the hierarchical structure of the semantic tree preserves semantic relationships. The tree grows and the session advances level-by-level with the progress of queries. Queries are compared and retrieved only within the context of the same parent node, ensuring accuracy while limiting the search space. This design allows sessions to share several initial queries at the beginning while enabling diverse and specific questions to branch out independently as conversations progress.

### 3.1.2 Semantic Node

Each semantic node represents the context of a single conversation turn. As shown in Figure 2, the embedding vector of the query, generated by an embedding model is used by the parent node for similarity search. Consequently, each semantic node stores the embedding vectors of its child nodes in a vector index [11, 24, 41, 9]. The KV cache for the associated context is maintained on the LLM side. To enable transparent KV cache sharing, an additional indirect KV cache block mapping table is introduced and managed by the corresponding semantic node, which is detailed in Section 3.3.

### 3.1.3 Semantic Forest

Multiple semantically individual trees collectively form the semantic forest. A global vector index stores the embedding vectors of the root nodes of all semantic trees, which is searched by the initial query of a new conversation session to locate the most relevant semantic tree. If no similar topic is found, a new semantic tree is created.

Users may occasionally change topics within a single session, such as shifting from one topic to another. In SmartCache, an active session can switch to another semantic tree if no similar queries are found under the current context and the new query is deemed irrelevant to the ongoing conversation.

To determine whether a query is contextual or not, SmartCache leverages the attention score from the LLM's query prefilling stage, enabling dynamic and accurate context switching.

---

**Algorithm 1** Cache Operations

---

**Require:** Session $s$, New Query $q_{new}$, Semantic Forest $\mathcal{F}$ (access to $I_{global}$ and local $I_{\mathcal{N}}$), Similarity threshold $\tau_{sim}$, Embedding Model $E(\cdot)$.

**Ensure:** Boolean $is\_hit$, Node $N_{hit}$, Response $a_{cached}$

1: **function** CACHELOOKUP($s, q_{new}, \mathcal{F}, \tau_{sim}$)
2:     $N_{cur} \leftarrow CurrentNode(s)$
3:     $e_{new} \leftarrow E(q_{new})$
4:     $is\_hit \leftarrow$ false
5:     $N_{hit} \leftarrow$ null
6:     $a_{cached} \leftarrow$ null
7:     **if** $N_{cur} = START$ **then**
8:         $Candidates \leftarrow NearestNeighbor(e_{new}, I_{global}, k = 1, \tau_{sim})$
9:         **if** $Candidates$ is not empty **then**
10:             $N_{hit} \leftarrow Candidates[0].Node$
11:             $a_{cached} \leftarrow N_{hit}.a$
12:             $is\_hit \leftarrow$ true
13:     **else**
14:         **if** $N_{cur}$ has a local index $I_{N_{cur}}$ **then**
15:             $Candidates \leftarrow NearestNeighbor(e_{new}, I_{N_{cur}}, k = 1, \tau_{sim})$
16:             **if** $Candidates$ is not empty **then**
17:                 $N_{hit} \leftarrow Candidates[0].Node$
18:                 $a_{cached} \leftarrow N_{hit}.a$
19:                 $is\_hit \leftarrow$ true
        **return** $is\_hit, N_{hit}, a_{cached}$

---

### 3.1.4 Cache Operations

SmartCache's cache operation is responsible for deciding whether an existing semantic node can be reused or whether the underlying LLM must be invoked to extend the semantic tree, whenever a new conversation turn arrives. The procedure (as shown in Algorithm 1) consists of three phases: *local lookup*, *global fallback* and *miss handling*.

• **Local Lookup.** Every session $s$ records its current semantic node $N_{cur}$. The search scope is constrained to the descendants of $N_{cur}$, which preserves the conversational continuity and bounds the search space. A single-probe $k = 1$ nearest-neighbor query on the local index $I_{N_{cur}}$ finds the most similar child with a minimum similarity of $\tau_{sim}$, using the embedding vector $e$ given by the embedding model. On cache hit, the cached response $a_{cached}$ of the hit node is streamed to the user and $N_{cur}$ is updated.

• **Global Fallback.** On failure of local lookup, SmartCache suspects a context shift. Before falling back to LLM generation, it performs a root-level lookup on the global vector index that stores the root nodes of semantic trees. A successful match moves the session to the hit root and returns cached response.

• **Miss Handling.** A double-miss triggers LLM inference and semantic node creation: a) *Attention Peeking.* After a single-pass prefilling, attention scores to historical tokens are inspected to determine a context-switch. If the query is contextual, the decoding proceeds. Otherwise, a new semantic tree is created and the generation is restarted without context. b) *Node Creation and KV Cache Materialization.* SmartCache inserts a new semantic node to an appropriate position with newly generated response $a_{new}$, and corresponding KV cache blocks, updating vector indices and setting up correct mappings.

### 3.1.5 Analysis and Discussion

Let $d$ be the embedding dimension of vector index, $|F|$ the number of semantic trees in the forest, and $|B|$ the average branching factor of a semantic node, $T$ the average generation length, and $|V|$ the vocabulary size.

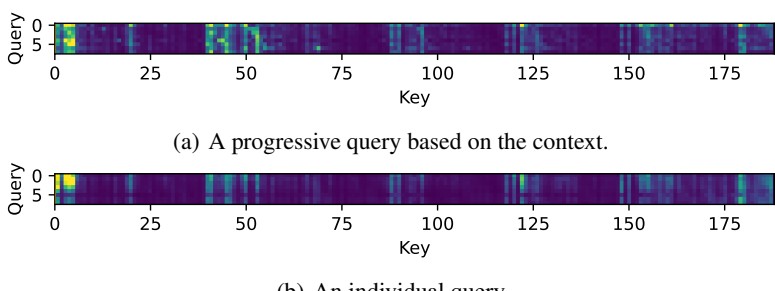

(a) A progressive query based on the context.

(b) An individual query

Figure 4: The attention score heatmaps of two different queries to the same context.

• **Asymptotic Time Cost.** SmartCache performs at least once approximate-nearest-neighbor [24] search over:

$$I_{global} : O(\log |F|) \text{ or,}$$
$$I_{local} : O(\log |B|)$$

• **Space Overhead.** For each semantic node, SmartCache stores a) a $d$-dimensional embedding vector ($4d$ Bytes in FP32 format), b) cached response $a_{cached}$ ($O(T \log |V|)$ Bytes) and c) local vector index $I_{local}$ consisting of index metadata ($O(\text{IndexSize}(|B|))$ Bytes) depending on the ANN implementation and the pointers to descendant embedding vectors ($8|B|$ Bytes). Therefore, the space overhead of each semantic node is:

$$O\left(d + T \log |V| + \text{IndexSize}(|B|) + |B|\right)$$

## 3.2 Dynamic Context-Switching

In multi-turn conversations, the sequence length of a user query is typically much shorter than that of the generated response. Additionally, prefilling a sequence is significantly faster than decoding the same number of tokens due to the batch processing nature of the prefilling stage. As a result, query prefilling consumes far less time than response generation. The attention score reveals the model's focus on key tokens, with contextual queries exhibiting noticeably higher attention scores to the context compared to individual queries, as illustrated in Figure 4. This observation motivates our approach of quickly examining the attention score during the query prefilling stage to determine whether a query is contextual. If the query is contextual, the LLM proceeds to decode the response. Otherwise, the query is treated as an individual query, and the session is redirected to another semantic tree. Since the attention score is already computed during prefilling, the performance overhead of this "peeking" mechanism is negligible.

By switching to a semantic tree with lighter contexts, sessions achieve better response times due to reduced computational load. Furthermore, the semantic forest expands with additional individual semantic contexts, increasing opportunities for sharing and improving overall efficiency.

## 3.3 KV Cache Management

### 3.3.1 Transparent Cross-Session KV Cache Sharing

SmartCache enables transparent sharing of KV cache across semantically similar conversation sessions by introducing a two-level mapping table. Original paged KV cache management relies on a per-session logical-to-physical token block mapping table, often referred to

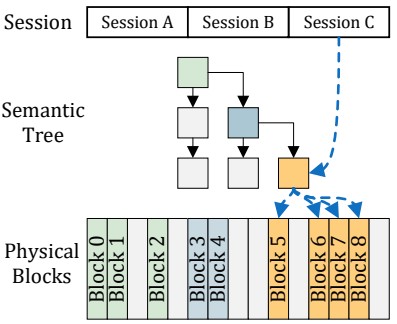

Figure 5: Two-Level Mapping Table.

as a block table. As illustrated in Figure 5, SmartCache enhances this approach with two additional structures: *per-session context tables*, which map logical tokens to contexts, and *per-context block*

*tables*, maintained by each semantic node. This two-level mapping mechanism ensures efficient and flexible KV cache sharing while preserving semantic relationships.

### 3.3.2 Semantic-aware Tiered Eviction/Prefetching

• **Semantic Tree-Guided Eviction.** SmartCache evaluates the eviction urgency $S_{evict}$ of a semantic node by two factors: *Semantic Depth Score (SDS)* and *Session Affinity Score (SAS)*. The SDS is calculated as $SDS = \frac{1}{\text{Depth}(N)}$, following the heuristic that deeper nodes in the semantic trees are prioritized to be evicted. The SAS, computed by $SAS = \frac{1}{\#\text{Session}(N)}$, implies that nodes shared across fewer active sessions should be evicted first. The weighted eviction score is as follows:

$$S_{evict} = \alpha \cdot SDS + \beta \cdot SAS$$

• **Subtree-aware Prefetching.** When a session advances to a semantic node $N$, SmartCache prefetches its children $C$ into the GPU memory based on the *Node Access Frequency (Popularity)* $AF(C_i) = \frac{\#\text{sessions accessed } C_i \text{ after } N}{\#\text{sessions reached N}}$. It is calculated using statistics within a predefined time window, given that hot topics may change over time.

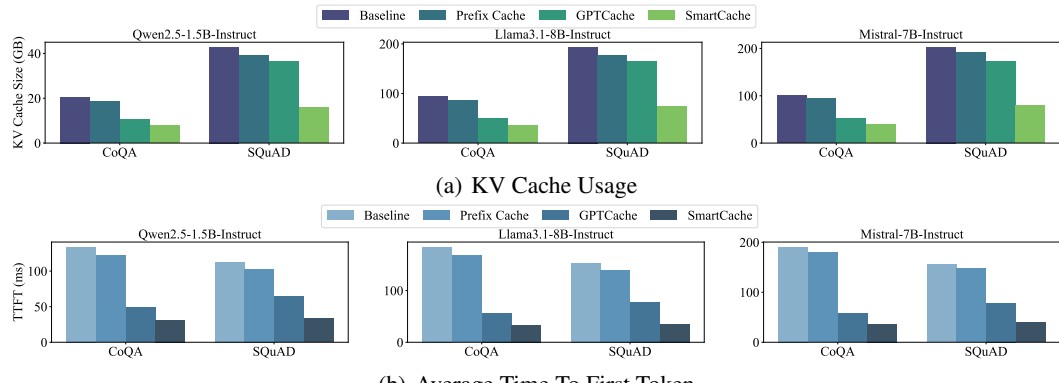

(a) KV Cache Usage

(b) Average Time To First Token

Figure 6: Resource efficiency comparison on different models and datasets.

## 4 Evaluation

### 4.1 Experiment Setup

**Harward settings.** We use a server equipped with Intel Xeon Silver 4310 CPU [17] and NVIDIA RTX4090 GPU [26], connected through PCIe4.0×16. The host has 256GB DDR4 memory and GPU has 24GB GDDR6X memory. Our server runs Linux 5.4 with CUDA 12.0 [5]. We implement our method using PyTorch 2.3 [29]. We evaluate our method on three different-sized open-source LLMs: Qwen-2.5-1.5B-Instruct [43], Llama-3.1-8B-Instruct [16], and Mistral-7B-Instruct-v0.2 [3].

**Dataset.** Three datasets are used in the evaluation, including the CoQA dev dataset [33] and SQuAD2.0 dev dataset [31]. CoQA is a conversational question answering dataset where questions are asked sequentially on the same short story. Later questions often depend on conversation history. SQuAD (Stanford Question Answering Dataset) contains questions on Wikipedia pages. Each story or paragraph used in the experiments has one original conversation sessions and two similar conversation sessions, with each session consisting of on average 5 turns of progressive question and answers.

**Methods.** Four methods are compared in the evaluation: 1) **Baseline**: KV caches are not shared among different conversations sessions, 2) **PrefixCache**: KV caches of common prefixes from different conversation sessions are shared, 3) **GPTCache**: an extra query-answer cache layer using the embeddings of queries as vector search keys is stacked on top of LLM inference backend, 4) **SmartCache**: the hierarchical and co-designed cache proposed in this paper. Conversation queries are issued interleavingly among different sessions.

The KV cache block size is 16 tokens. BGE-M3 [7] is used as the embedding model with the dimension of 1024. Embedding vectors are stored and searched using faiss [11] vector index based on L2 distance. The similarity threshold is set to 0.75.

Table 2: Answer Quality. Ideal represents the results of Baseline and PrefixCache.

| Benchmark | | CoQA (0-shot) | | | SQuAD2.0 (0-shot) | | |
|---|---|---|---|---|---|---|---|
| Model | Method | ExactMatch | F1 | ROUGE-L | ExactMatch | F1 | ROUGE-L |
| Qwen-2.5-1.5B-Instruct | Ideal | 27.5 | 51.1 | 51.3 | 14.2 | 65.5 | 64.0 |
| | GPTCache | 16.8 | 33.3 | 33.8 | 12.8 | 60.4 | 59.2 |
| | SmartCache | **24.9 (+48.2%)** | **46.6 (+39.9%)** | **47.0 (+39.1%)** | **13.4 (+4.7%)** | **64.5 (+6.8%)** | **63.0 (+6.4%)** |
| Llama-3.1-8B-Instruct | Ideal | 36.6 | 59.6 | 59.2 | 14.9 | 67.8 | 66.0 |
| | GPTCache | 23.0 | 38.8 | 39.1 | 13.4 | 62.3 | 60.9 |
| | SmartCache | **24.9 (+8.3%)** | **49.5 (+20.1%)** | **49.2 (+20.2%)** | **14.0 (+4.5%)** | **66.5 (+6.7%)** | **64.6 (+6.1%)** |
| Mistral-7B-Instruct | Ideal | 5.5 | 27.0 | 26.9 | 7.7 | 57.8 | 55.3 |
| | GPTCache | 3.0 | 17.4 | 17.7 | 6.9 | 52.7 | 50.6 |
| | SmartCache | **3.1 (+3.3%)** | **20.0 (+14.9%)** | **20.0 (+13.0%)** | **7.7 (+11.6%)** | **56.2 (+6.6%)** | **54.6 (+7.9%)** |

## 4.2 Resource Efficiency

**Memory Efficiency.** SmartCache significantly improves KV cache sharing by leveraging semantic relationships in multi-turn conversations, enhancing resource utilization across sessions. As shown in Figure 6(a), SmartCache reduces KV cache memory usage by up to $59.1\%$ and $56.0\%$ compared to PrefixCache and GPTCache, respectively, demonstrating superior efficiency in reusing KV cache memory.

**Computational Efficiency.** Figure 6(b) presents the average Time to First Token (TTFT) per query. On the CoQA dataset, across different models, SmartCache achieves $78.0\%$ and $37.7\%$ lower TTFT than PrefixCache and GPTCache, respectively. Similarly, on the SQuAD dataset, it reduces TTFT by $71.7\%$ and $50.6\%$ compared to the same baselines. These results highlight SmartCache's effectiveness in accelerating inference while minimizing computational overhead.

## 4.3 Answer Quality

The hierarchical semantic awareness of SmartCache significantly improves its answer quality, compared with GPTCache's flat query similarity search. As demonstrated in Table 2, SmartCache consistently enhances answer quality across different benchmarks and metrics. For the CoQA dataset, SmartCache improves Qwen-2.5-1.5B-Instruct's F1 score by $39.9\%$ and Llama-3.1-8B-Instruct's ROUGE-L by $20.2\%$. For SQuAD2.0, SmartCache achieves smaller but consistent gains.

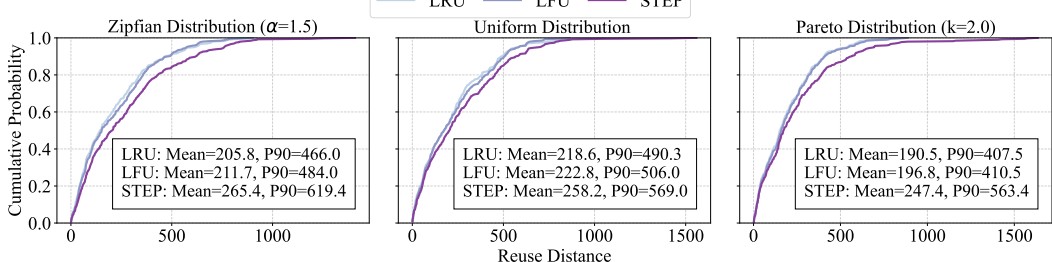

Figure 7: Reuse Distances CDF of different KV Cache Management Policy

## 4.4 KV Cache Management Policy

We compare SmartCache's STEP policy with commonly used LRU (Least Recently Used) and LFU (Least Frequently Used) policies. 1500 conversation sessions focusing on CoQA's stories follow three different distributions (zipfian, uniform and pareto) to show different real-world environments. Llama-3.1-8B-Instruct model is used with a GPU KV Cache budget of 4GB. Figure 7 compares the CDF (Cumulative Distribution Function) of the Reuse Distances (the number of distinct nodes accessed between two consecutive accesses to the same semantic node) of different policies. STEP shows up to $29.9\%$ and $25.7\%$ longer average reuse distance over LRU and LFU with skewed distribution and $18.2\%$ and $15.9\%$ longer with uniform distribution.

## 4.5 Overhead Analysis

We quantitatively show SmartCache's compute and memory overhead on Mistral-7B-Instruct-v0.2 using CoQA. Table 3 demonstrates the breakdown of overhead across SmartCache components. For a cache-missed new query (i.e. creating a new semantic node followed by LLM inference), Smart-Cache adds $\approx 8.8\%$ computational overhead (18.54 ms out of 210.37 ms end-to-end). SmartCache's

hierarchical semantic organization and co-design with KV cache management increase cache reuse opportunities, leading to significant end-to-end gains despite modest overhead.

The total GPU memory overhead is ≈7.1%, primarily from the embedding model, while CPU memory overhead is negligible (≈28.05 MB). Given the reduced inference time from cache reuse, the amortized cost of these components becomes insignificant at scale.

### 4.6 Performance under High Concurrency

As shown in Figure 8, we evaluate SmartCache under varying degrees of concurrent requests using Mistral-7B-Instruct-v0.2 on CoQA. As the number of concurrent requests increases, SmartCache demonstrates progressively greater speedups. As concurrency increases, the LLM's batching efficiency plateaus due to inherent GPU parallelism limits. However, SmartCache is able to bypass full LLM inference for many cache-hit queries, reducing the effective batch size. This not only alleviates load on the inference engine but also leverages hierarchical KV cache reuse, leading to significant reductions in TTFT.

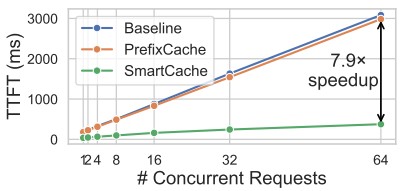

Figure 8: Overhead under different number of concurrent requests.

Table 3: Computational and Memory Overhead Breakdown of SmartCache.

| Components | Computational Overhead | Memory Overhead |
|---|---|---|
| Embedding Generation | 17.13 ms | 1.06 GB (GPU) |
| Vector Search | 0.11 ms | 18.16 MB (CPU) |
| Vector Insertion | 1.01 ms | Shared with vector search |
| Semantic Forest Maintenance | 0.29 ms | 9.89 MB (CPU) |

## 5 Related Works

PrefixCache [40] reuses KV cache only when token prefixes match exactly, accelerating prefill but is limited in handling semantically similar queries. GPTCache [6] stores and retrieves responses based on flat query embeddings, enabling semantic reuse but lacking context awareness. PromptCache [15] accelerates inference by reusing attention (KV) states for frequently repeated prompt segments via a modular schema that preserves positional consistency, substantially reducing TTFT for long, template-like inputs. CacheBlend [44] targets RAG workloads, enabling chunk-level KV cache reuse across retrieved documents and selective recomputation of a small subset of tokens to refresh cross-chunk attention without incurring full prefill cost. Notably, SmartCache can be combined with CacheBlend and PromptCache to achieve further improvements.

## 6 Conclusion and Discussion

In this paper, we presented SmartCache, a context-aware semantic cache framework designed to address the inefficiencies of multi-turn LLM inference. By integrating hierarchical semantic indexing with KV cache management, SmartCache significantly reduces redundant computations and GPU memory usage while preserving conversational context.

**Limitations and Future Work.** First, sharing responses across user sessions introduces potential privacy risks. To mitigate this, a content desensitization module could be implemented to automatically remove or anonymize sensitive information prior to caching. Second, scaling the system to support a broader range of services requires a multi-instance extension, enabling more robust and distributed processing capabilities.

## Acknowledgments and Disclosure of Funding

The work described in this paper is partially supported by the grants from the Research Grants Council of the Hong Kong Special Administrative Region, China (GRF 14202123, GRF 14200224).

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

# A Appendix

## A.1 Evaluation on Larger Model

To further evaluate SmartCache's scalability and effectiveness, we conduct additional experiments using the Qwen2.5-32B-Instruct model on a high-end hardware configuration: AMD EPYC 7513, 512GB DRAM, and one NVIDIA A100 80GB GPU. Table 4 summarizes the results on the CoQA dataset.

Table 4: Results on Qwen2.5-32B-Instruct

| Metrics / Method | TTFT (ms) | KV Cache (GB) | EM | F1 | ROUGE-L |
|---|---|---|---|---|---|
| Baseline | 213.9 | 189.2 | 33.3 | 54.8 | 55.0 |
| PrefixCache | 182.5 | 171.8 | 33.3 | 54.8 | 55.0 |
| GPTCache | 94.1 | 98.3 | 18.8 | 34.3 | 34.7 |
| SmartCache | **69.4** | **71.4** | 26.8 | 50.2 | 50.4 |

## A.2 Computational Overhead under Concurrency

We evaluate SmartCache's computational overhead using Mistral-7B-Instruct-v0.2 on CoQA under varying numbers of concurrent requests. As shown in Table 5, the primary cost under increased concurrency arises from embedding generation, which supports efficient batching. 8 CPU threads are used for faiss-cpu internal threading and semantic forest maintenance. Minor contention is observed in the semantic forest maintenance, especially when multiple queries attempt to insert child nodes under the same parent node. This is mitigated by batching vector insertions.

Table 5: Overhead under concurrent requests (ms).

| # Concurrent Requests | 1 | 2 | 4 | 8 | 16 | 32 | 64 |
|---|---|---|---|---|---|---|---|
| Embedding Generation | 17.13 | 18.00 | 19.44 | 19.54 | 20.81 | 22.08 | 23.32 |
| Vector Search | 0.11 | 0.23 | 0.26 | 0.36 | 0.75 | 5.64 | 9.35 |
| Vector Insertion | 1.01 | 1.01 | 1.02 | 1.05 | 1.06 | 1.22 | 1.62 |
| Semantic Forest Maintenance | 0.29 | 0.31 | 0.31 | 0.33 | 0.67 | 1.34 | 2.97 |

## A.3 Hyperparameter Sensitivity

We evaluate similarity threshold $\tau_{sim}$ used to determine semantic matches in the Semantic Forest, and the weighting coefficients $(\alpha, \beta)$ that balance the Semantic Depth Score (SDS) and Session Affinity Score (SAS) in the STEP eviction policy.

### A.3.1 Similarity Threshold

Table 6 shows the answer quality and performance on Qwen2.5-1.5B-Instruct with CoQA. Increasing $\tau_{sim}$ monotonically improves EM/F1/ROUGE-L by enforcing stricter semantic matches, but reduces reuse opportunities, increasing both TTFT and KV cache usage. We adopt $\tau_{sim} = 0.75$ for a balanced operating point used in main experiments.

Table 6: Answer quality and performance across different similarity threshold.

| $\tau_{sim}$ | EM | F1 | ROUGE-L | TTFT (ms) | KV Cache (GB) |
|---|---|---|---|---|---|
| 0.50 | 17.4 | 35.2 | 35.5 | 26.8 | 4.1 |
| 0.60 | 21.6 | 41.1 | 41.4 | 27.5 | 4.5 |
| 0.70 | 23.5 | 45.3 | 45.5 | 30.2 | 6.5 |
| 0.75 | 24.9 | 46.6 | 47.0 | 31.3 | 7.7 |
| 0.80 | 25.4 | 47.7 | 47.9 | 49.0 | 8.5 |
| 0.90 | 25.9 | 48.9 | 49.1 | 83.2 | 12.1 |

### A.3.2 Coefficients $\alpha$ and $\beta$ of STEP Policy

Table 7 shows the grid search result with the same configuration as Section 4.4. We adopt $\alpha = 0.6$ and $\beta = 0.4$ for STEP in all evaluations, as this setting attains the highest average reuse distance.

Table 7: Average reuse distance over different $(\alpha, \beta)$ settings (zipfian distribution).

| $(\alpha, \beta)$ | (0.1,0.9) | (0.2,0.8) | (0.3,0.7) | (0.4,0.6) | (0.5,0.5) | (0.6,0.4) | (0.7,0.3) | (0.8,0.2) | (0.9,0.1) |
|---|---|---|---|---|---|---|---|---|---|
| Reuse distance (avg) | 231.8 | 252.0 | 255.9 | 262.3 | 261.6 | **265.4** | 260.5 | 240.9 | 252.5 |

### A.4 Attention-based Context Switching

Since different models exhibit numeric variations on attention scores, the attention score threshold is profiled for each model offline using five predefined independent queries, averaged across attention heads and query tokens for each query, excluding 8 initial tokens that behave as attention sinks (i.e., attracting disproportionate attention regardless of context [42]). The same attention score calculation procedure is triggered during the prefill stage of a query when it fails from reusing its child semantic nodes. It is then compared with the the profiled threshold to determine whether or not the query is independent.

**Impact of Architectural Differences.** The widely used Grouped-Query Attention (GQA [2]) changes per-head magnitudes but preserves the probabilistic semantics of attention. Mixture-of-Experts (MoE [13]) alters the feed-forward pathway via expert routing while leaving the attention softmax itself unchanged. Positional encodings such as rotary position embeddings (RoPE [35]) and attention with linear biases (ALiBi [28]) modify how relative position influences attention logits, but after softmax the attention weights still form a probability simplex. Therefore, architectural modifications that change logits but preserve the probabilistic semantics do not require changing the decision logic.

### A.5 Privacy Preserving

SmartCache is designed to support multi-user environments, where multiple sessions may reuse cached responses across conversations. While semantic reuse offers significant performance and memory advantages, it also introduces potential privacy concerns. These issues can be mitigated via query sanitization, access control, and asynchronous privacy filtering.

**Query Sanitization.** To prevent private or sensitive information from being inadvertently shared across sessions, SmartCache integrates a privacy data sanitizer module. This module inspects user queries and determines whether a query-response pair should be treated as private. If so, the corresponding semantic node is marked as hidden, meaning it is only visible to the originating user or authorized parties, such as members of the same organization or tenant.

**Access Control.** Hidden semantic nodes are excluded from the global semantic forest used for cross-session lookup. These nodes are stored and reused exclusively within the user's own session or within authorized scopes defined by privacy policies. This ensures that personal or sensitive queries, even if semantically similar to others, do not produce cache hits across different users.

**Asynchronous Sanitization for Low Latency.** To minimize latency, when a new query is issued, it is immediately processed and added to the cache as a hidden node by default. In the background, the sanitizer evaluates the query and, if it is deemed non-sensitive, updates the node's visibility status to allow future sharing. This decouples privacy inspection from the critical path of inference, preserving the low TTFT guarantees of SmartCache while still ensuring privacy compliance.

