# OpenReview forum: "SmartCache: Context-aware Semantic Cache for Efficient Multi-turn LLM Inference"
_NeurIPS.cc/2025/Conference — NeurIPS 2025 poster_

### Official Review · Reviewer_MXSA · 2025-06-23

**Clarity:** 4
**Significance:** 3
**Originality:** 3
**Rating:** 5
**Confidence:** 4

**Summary:**

This paper introduces a novel method to store semantically equivalent queries and responses to reduce computation. While previous methods like Prefix caching only work for exact match queries, SmartCache searches for queries based on embedding similarity. The approach also takes history across multiple sessions into account, based on the observation that users typically ask several questions on the same topic, and so the method is able to more efficiently search a narrower space of cached queries based on the initial query. It also has a mechanism to detect context switches if the user changes topic.

**Questions:**

In the experimental design, did you experiment with different similarity thresholds for the embeddings? This seems to me to be a crucial parameter.

I am surprised that answer quality increased significantly using your method. I would have expected the baseline approach to perform best, i.e. when the model is free to generate an answer to every question, and that SmartCache would have sometimes mistakenly presented a cached answer when the similarity was high.

**Ethical Concerns:**

["NO or VERY MINOR ethics concerns only"]

**Final Justification:**

The main limitation of this paper was that the experimental design was crammed into the final 1.5 pages so a bunch of important experimental details were not described/analysed in sufficient detail.

The authors successfully responded to this criticism with lots of further results tables addressing the questions from each reviewer, which gives me confidence that the experiments are robust. The authors should work these results/discussions into the camera ready version of the paper or the appendix.

**Paper Formatting Concerns:**

The authors did not provide any justifications in the Paper Checklist.

Many references are missing the journal/conference venue; e.g. Chen et al. is from ACL 2024.

**Quality:**

2

**Strengths And Weaknesses:**

Strengths:

The work is well-motivated, and the solution is logical and well-explained.
The paper is very clear and comprehensive in describing all the components of the solution and walking the reader through it – it was a pleasure to read!
Experiments are carried out on 3 different LLM families and show gains across all of them.

Weakness:

It feels like the authors ran out of space, and so the experimental descriptions/analysis is not detailed enough. The whole experimental setup, evaluation, analysis and conclusion are crammed into the final 1.5 pages. I still think the authors did just enough to make the paper publishable, but would encourage them to expand this section and move some earlier sections into an appendix.

---

> ### Author Rebuttal · Authors · 2025-07-31
>
> We appreciate your acknowledgment of our work as **well-motivated** and the paper as **well-explained**. Below, we provide detailed responses to your constructive comments.
>
> **Q1: Different Similarity Threshold**
>
> **Response:** Thank you for your valuable comment. The similarity threshold τ_sim is set to 0.75 to give a balanced trade-off between performance and quality in the evaluation. Table 1 below shows the ablation result of τ_sim with Qwen2.5-1.5B-Instruct on CoQA.
>
> Table 1: Different similarity threshold.
>
> | Threshold | EM   | F1   | ROUGE-L | TTFT (ms) | KV Cache (GB) |
> |-----------|------|------|:--------|:----------|:--------------|
> | 0.5       | 17.4 | 35.2 | 35.5    | 26.8      | 4.1           |
> | 0.6       | 21.6 | 41.1 | 41.4    | 27.5      | 4.5           |
> | 0.7       | 23.5 | 45.3 | 45.5    | 30.2      | 6.5           |
> | 0.75      | 24.9 | 46.6 | 47.0    | 31.3      | 7.7           |
> | 0.8       | 25.4 | 47.7 | 47.9    | 49.0      | 8.5           |
> | 0.9       | 25.9 | 48.9 | 49.1    | 83.2      | 12.1          |
>
> By increasing τ_sim, answer quality can be further improved. However it leads to higher TTFT and increased KV cache usage resulted from reduced opportunities to reuse similar queries.
>
> **Q2:** I am surprised that answer quality increased significantly using your method. I would have expected the baseline approach to perform best, i.e. when the model is free to generate an answer to every question, and that SmartCache would have sometimes mistakenly presented a cached answer when the similarity was high.
>
> **Response:** Thank you for the comment. To clarify, as shown in Section 4.3 (Table 2), the answer quality of SmartCache is close to the ideal and better than GPTCache. Additionally, as shown above, there is a trade-off between the performance gains and slight quality degradation compared to the ideal.
>
> **Q3:References missing conference/journal venue**
>
> **Response:** Thank you for pointing out the missing conference and journal venues in the references. We will correct the formatting in the revised version.

---

> ### Comment · Reviewer_MXSA · 2025-08-05
>
> Thank you for the results showing how the value of the similarity threshold affects performance, speed, and KV Cache usage.
> These results are reassuring and should absolutely be included in the final paper.
>
> The detailed rebuttals to other reviewer questions, supported by experimental results, are also reassuring and should likewise be included in the camera ready version of the paper or the appendix.
>
> I am satisfied the authors have done enough to warrant acceptance at the conference, and so have increased my score accordingly.

---

> > ### Author Response · Authors · 2025-08-05
> >
> > Thank you very much for your thoughtful feedback and encouragement. We sincerely appreciate your recognition of our efforts to address the reviewer concerns with detailed experiments and clarifications. We will ensure that the analysis of the similarity threshold, as well as the detailed responses to the reviewer questions, are included in the camera-ready version of the paper or the appendix as appropriate.

---

### Official Review · Reviewer_P9MM · 2025-06-29

**Clarity:** 3
**Significance:** 3
**Originality:** 3
**Rating:** 4
**Confidence:** 3

**Summary:**

This paper proposes Smart Cache, a system level co-design framework to improve the efficiencies of multi-turn conversations that reduces the TTFT and kv cache useage. This method first cluster the conversation histories into the topic-specific tree “Semantic Forest,” performs a local-then-global ANN search over saved embeddings, and then dynamically decide whether to start a new tree or stays in the current tree based on the attention during prefill. The paper also proposes STEP policy to decide the eviction/ prefetch based on the two-level mapping table. Experiments on CoQA and SQuAD with Llama-3-8B demonstrate up to 59 % KV-memory reduction and 78 % faster time-to-first-token versus prefix caching.

**Questions:**

Have you examined how large the semantic tree would be confronting large volume of requests?

What happens if the local lookup fails, but a semantically very similar node exists elsewhere in the same tree (i.e., not as a direct child)?

Could you report the efficiency gains under minimal performance losses? People won’t like significant compromise of performance even with significant TTFT gains

**Ethical Concerns:**

["NO or VERY MINOR ethics concerns only"]

**Final Justification:**

My main concern is in the lack of the experiments part and the lack of the breakdown parts. The author has provided additional experiments in the rebuttal and a clear breakdown of the overhead. While I raise the score from 3 to 4, I hope more efforts could be done for discussing the trade-off, since we can still see the performance degradation.

**Limitations:**

I think the main limitation is for the experiment sessions. Please see my comments in the weakness parts.

**Quality:**

2

**Strengths And Weaknesses:**

**Strength**
- The author discusses on an important topic that expand the idea of GPT-cache to the multi-turn conversations and is able to reduce both the kv cache memory and the time-to-first token
- Compared to the baselines outlined in the paper, this method achieves better performance than GPT-cache
- The author proposes system-algorithm co-design that enables efficient kv-cache reuses

**Weakness**
- Limited Scale Evaluation: The evaluation is conducted on relatively small models and datasets. It's unclear how the approach would scale to larger models (e.g., 32B+ parameters) or handle a significant volume of concurrent requests. Some evaluations on larger models, or other large multi-turn conversations, such as Lmsys-chat, would provide a more comprehensive evaluation of this method.
- Response quality degradation: As shown in the Table2, even with the smart cache, the performance on the CoQA is still significantly lower than the ideal one.
- Privacy concerns: Though the authors have noted the privacy limitations in the paper, no concrete solutions have been proposed to address this concern.
- Lack of ablation and breakdown experiments: This paper doesn’t provide enough analysis on how the overhead of the retrieval, possible loading time during PCIE, and ANN search. When the prefix is short and the model is small, these overheads could be significant.
- Lack of baselines: More recent paper, such as CacheBlend and PromptCache, also target on the efficient caching for reducing TTFT. These baselines are not included.

---

> ### Author Rebuttal · Authors · 2025-07-31
>
> Thank you for ackownledging the **important topic** addressed in our work. Your detailed questions are constructive and valuable to us. Below, we provide detailed responses to your comments.
>
> **Q1: Larger Scale**
>
> **Response:** Thank you for the comment. Experimental results with a larger model Qwen2.5-32B-Instruct on AMD EPYC 7513, 512GB DRAM equipped with one NVIDIA A100 80GB are shown as follows.
>
> Table 1: Results of Qwen2.5-32B-Instruct on CoQA
>
> | Metrics/Methods | TTFT (ms) | KV Cache (GB) | EM   | F1   | ROUGE-L |
> |-----------------|-----------|---------------|:-----|:-----|:--------|
> | Baseline        | 213.9     | 189.2         | 33.3 | 54.8 | 55.0    |
> | PrefixCache     | 182.5     | 171.8         | 33.3 | 54.8 | 55.0    |
> | GPTCache        | 94.1      | 98.3          | 18.8 | 34.3 | 34.7    |
> | SmartCache      | 69.4      | 71.4          | 26.8 | 50.2 | 50.4    |
>
> The results on a large dataset such as Lmsys-chat will be included in the revised version.
>
> **Q2: Efficiency Gains w.r.t Quality Loss** Could you report the efficiency gains under minimal performance losses? People won’t like significant compromise of performance even with significant TTFT gains
>
> **Response:** Thank you for the comment. Table 2 shows the answer quality and performance of SmartCache when using a high similarity threshold of 0.9.
>
> Table 2: Quality v.s. Performance
>
> |                          | EM   | F1   | ROUGE-L | TTFT (ms) |
> |--------------------------|------|------|:--------|:----------|
> | Qwen2.5-1.5B-Instruct    | 25.9 | 48.9 | 49.1    | 83.2      |
> | Llama3.1-8B-Instruct     | 30.3 | 54.7 | 54.5    | 101.7     |
> | Mistral-7B-Instruct-v0.2 | 4.6  | 23.8 | 23.7    | 118.2     |
>
> The quality is compromised by little with a high similarity threshold.
>
> **Q3:** What happens if the local lookup fails, but a semantically very similar node exists elsewhere in the same tree (i.e., not as a direct child)?
>
> **Response:** Thank you for the in-depth comments. In such cases, SmartCache treats nodes in different branches as semantically different. Since they originate from different parent contexts, forcing a cache hit across branches could result in false positives and potentially incorrect answers. We have considered ways to accurately handle such cases, recognizing that a more complex data structure and algorithm would be required. After evaluating the trade-off between complexity and speed, we chose to allow a locally cache-missed query to either start a new tree or create a new child node under the current node.
>
> **Q4: Privacy Concerns**
>
> **Response:** Thank you for the insightful comment. The solution to the privacy issue will be clarified in the revised version. Specifically, SmartCache can incorporate a user privacy data sanitizer module to check whether the queries contain private information. Private query-response semantic nodes are hidden from other users. Only the owner or users within the same organization can access the hidden nodes, based on the privacy configurations. Various existing privacy sanitization methods can be employed as the sanitizer. To minimize potential overhead and latency cause by the privacy check, a new query can first be processed by SmartCache and inserted as an hidden semantic node. It will then be asynchronously examined by the background sanitizer to determine whether to release its hidden status.
>
> **Q5: Ablation and Breakdown Evaluations**
>
> **Response:** Thank you for the comment. As shown in Table 1, the computational and memory overhead with Mistral-7B-Instruct-v0.2 on CoQA. The computational overhead for processing a cache-missed new query (i.e. setting up a new semantic node + performing LLM inference) is approximately 8.8% (18.54ms out of 210.37ms). Due to SmartCache's hierarchical semantic information and co-design with KV cache management, there is a higher likelyhood of reusing cached responses resulting in more substantial performance gains. SmartCache introduces about 7.1% GPU memory overhead (embedding model compared to LLM weights and compute buffer) and little CPU memory overhead (28.05MB).
>
> Table 3: Overhead Breakdown
>
> | Breakdown                    | Computational Overhead | Memory Overhead           |
> |------------------------------|------------------------|---------------------------|
> | Embedding Generation         | 17.13ms                | 1.06GB GPU memory         |
> | Vector Search                | 0.11ms                 | 18.16MB CPU memory        |
> | Vector Insertion             | 1.01ms                 | Shared with vector search |
> | Semantic Forest Maintenance | 0.29ms                 | 9.89MB CPU memory         |
>
>
> **Q6: Comparison with More Baselines**
>
> **Response:** Thank you for the comment. Those works will be included in the revised version. Specifically, CacheBlend aims at speeding up the KV cache computation of different document chunks in RAG scenario. PromptCache aims at efficiently reusing the attention states like KV cache of frequently repeated input prompts, such as system prompts. SmartCache is designed to identify and reduce computational and memory overhead for similar queries in multi-user multi-turn conversations scenario. Notably, SmartCache can be combined with CacheBlend and PromptCache to achieve further improvements.

---

> > ### Comment · Reviewer_P9MM · 2025-08-03
> >
> > Thanks for the discussion. Your rebuttal addressed my concerns for the lack of experiments part. I will raise my score for making the additional experiments and the clarification for the breakdown of the overhead.
> > I notice in the table 1, smartcache still has the performance loss compared to the baseline and the prefix cache. I think more discussions regarding the tradeoff of the performance and the efficiencies should be discussed in the camera-ready version.

---

> > > ### Author Response · Authors · 2025-08-03
> > >
> > > Thank you for your insightful feedback and for recognizing the additional experiments and clarifications we have included. We sincerely appreciate your constructive suggestions to enhance our paper. In the camera-ready version, we will be sure to incorporate a more in-depth discussion of the tradeoff between performance and efficiency.

---

### Official Review · Reviewer_C7su · 2025-06-30

**Clarity:** 2
**Significance:** 2
**Originality:** 3
**Rating:** 5
**Confidence:** 3

**Summary:**

This paper proposes SmartCache, a novel system-algorithm co-design framework to address the inefficiencies of multi-turn LLM inference. SmartCache introduces a "Semantic Forest" structure to hierarchically index conversational turns, enabling efficient retrieval and reuse of responses while preserving conversational context. It further leverages internal LLM attention scores for dynamic context detection and introduces a "two-level mapping" for cross-session Key-Value (KV) cache sharing, complemented by a "Semantic-aware Tiered Eviction Policy (STEP)". The evaluation demonstrates substantial reductions in KV cache memory usage and Time-to-First-Token (TTFT), alongside improvements in answer quality on various benchmarks.

**Questions:**

Given that the paper mentions $t_{sim}$ for embedding similarity, could the authors clarify if a similar explicit threshold or a specific methodology for dynamically determining context changes based on LLM attention scores is employed in SmartCache, and how its sensitivity and generalizability across different LLM architectures and datasets were evaluated?

**Ethical Concerns:**

["NO or VERY MINOR ethics concerns only"]

**Final Justification:**

The rebuttal provides more experiment results. The idea is novel and the paper is well-organized.

**Limitations:**

Yes

**Quality:**

3

**Strengths And Weaknesses:**

Strengths:

- Introduces a hierarchical semantic cache (Semantic Forest) that respects conversational context. Bridges the gap between semantic and KV caching by linking semantic nodes directly to KV cache blocks, which is underexplored. Integrates attention-based context detection into the caching pipeline to mitigate misuses due to context-ignorant semantic retrieval.
- Demonstrates consistent improvements on CoQA and SQuAD benchmarks using three LLMs. Reduces KV cache usage by up to 59.1%, TTFT by up to 78.0%, and improves F1/ROUGE by up to 39%.
- The two-level KV cache mapping (session → semantic node → block) is a practical innovation that enables transparent sharing without compromising cache coherence or semantic fidelity.
- The paper is well-organized with clear diagrams and pseudocode, easy to understand.

Weaknesses
- While the Semantic Forest provides clear benefits, the operational overhead of maintaining and updating this complex hierarchical structure in a highly dynamic, real-time serving environment could be further elaborated, especially concerning potential locking mechanisms or distributed concerns for very large-scale deployments.
- The threshold for determining context changes based on attention scores might require fine-tuning across different LLM architectures or datasets. A more detailed analysis of its robustness would be beneficial.

---

> ### Author Rebuttal · Authors · 2025-07-31
>
> We appreciate your support of our work and your detailed detailed summary of the problem we address and novelty of our approach. Your comments are insightful and constructive to us.
>
> **Q1: Operational Overhead In Large-Scale**
>
> **Response:** Thank you for the valuable comment. The scalability of semantic forest under high concurrency is important for user experience in large-scale deployment. Table 1 shows the operational overhead of SmartCache in the scenario with varying numbers of concurrent queries, based on Mistral-7B-Instruct-v0.2 and CoQA.
>
> Table 1: Overhead analysis under different number of concurrent requests
>
> | # Concurrent Request        | 1     | 2     | 4     | 8     | 16    | 32    | 64    |
> |-----------------------------|-------|-------|:------|:------|:------|:------|:------|
> | Embedding Generation        | 17.13 | 18.00 | 19.44 | 19.54 | 20.81 | 22.08 | 23.32 |
> | Vector Search               | 0.11  | 0.23  | 0.26  | 0.36  | 0.75  | 5.64  | 9.35  |
> | Vector Insersion            | 1.01  | 1.01  | 1.02  | 1.05  | 1.06  | 1.22  | 1.62  |
> | Semantic Forest Maintenance | 0.29  | 0.31  | 0.31  | 0.33  | 0.67  | 1.34  | 2.97  |
>
> The primary source of overhead, embedding generation, supports efficient batching. 8 CPU threads are used for faiss-cpu internal threading and semantic forest maintenance. Contention mainly occurs when multiple queries attempt to add new child nodes to the same parent semantic node, which is mitigated by batching vector insertion.
>
> **Q2: More Analysis of Attention Score Threshold on Different LLM Architectures**
>
> **Response:** Thank you for the comment. We will incorporate a more detailed analysis of different models in the revised version.

---

> > ### Comment · Area_Chair_V4fM · 2025-08-05
> >
> > As the author-reviewer discussion will end on Aug 8, it would be great to leave your further comments for the authors' reply, or leave Mandatory Acknowledgement with your rating.

---

> > ### Comment · Reviewer_C7su · 2025-08-09
> >
> > Thank you for the detailed clarifications and additional data. The overhead breakdown in Table 1, along with the explanation of contention and batching strategies, addresses my scalability concern. I appreciate your commitment to adding the attention-threshold analysis across LLM architectures, which will strengthen the paper’s empirical depth. I think my original rating is reasonable. Thanks.

---

> > > ### Author Response · Authors · 2025-08-09
> > >
> > > Thank you for your feedback. We are glad that our experiments and explanations address your scalability concern, and we will include more attention-threshold analysis across different LLM architectures in the camera-ready version. We greatly appreciate your support.

---

### Official Review · Reviewer_h8B6 · 2025-07-02

**Clarity:** 3
**Significance:** 3
**Originality:** 3
**Rating:** 5
**Confidence:** 4

**Summary:**

This paper introduces SmartCache, a system-algorithm co-design framework aimed at improving multi-turn LLM inference efficiency by addressing redundant computation and memory usage caused by semantically similar queries across sessions. Unlike methods that require exact matches or ignore conversational context, SmartCache integrates context-aware semantic caching with the inference engine. Its key contributions include a hierarchical semantic forest for accurate reuse of responses, an attention-based module for detecting topic shifts, and an integrated KV cache management system with cross-session sharing and a novel STEP eviction policy. Experiments show that SmartCache significantly reduces KV cache usage and TTFT while improving answer quality.

**Questions:**

1. Overhead Analysis: Can the authors provide quantitative measurements on the computational and memory overhead introduced by embedding generation, vector search, and semantic forest operations?
2. Attention-based Context Switching: Could you elaborate on how the system distinguishes between "progressive" and "individual" queries? Is there a fixed threshold on attention scores, and how robust is this heuristic across different models?
3. Performance under High Concurrency: Have you evaluated SmartCache under high concurrency or batch inference settings? Since cross-session reuse is central to the system, understanding its behavior under realistic serving loads would be valuable.

**Ethical Concerns:**

["NO or VERY MINOR ethics concerns only"]

**Final Justification:**

SmartCache addresses a significant inefficiency in multi-turn LLM serving with a novel system-algorithm co-design that integrates semantic caching and KV cache management. My initial concerns focused on overhead quantification, scalability under high concurrency, and hyperparameter sensitivity. The authors’ rebuttal provided detailed breakdowns and new experiments demonstrating SmartCache’s additional overhead is minimal and its performance scales well with concurrency. The ablation and sensitivity analyses further strengthened the empirical evaluation. Given that the major concerns have been satisfactorily addressed, I believe the paper makes a solid technical contribution and have raised my rating from 4 to 5 and confidence from 3 to 4.

**Limitations:**

Yes.

**Paper Formatting Concerns:**

No.

**Quality:**

4

**Strengths And Weaknesses:**

### Strengths
1. High Significance: The paper tackles a crucial problem in LLM serving—inefficiency in multi-turn conversations caused by semantic redundancy.
2. Novel and Holistic Design: The integration of semantic caching and KV cache management reflects a thoughtful system-algorithm co-design, offering a comprehensive solution that surpasses prior approaches in both scope and capability.
3. Strong Empirical Results: SmartCache outperforms baseline, prefix, and flat semantic caching methods in terms of memory efficiency, TTFT, and answer quality across multiple models and datasets.

### Weaknesses
1. Lack of Overhead Quantification: While the authors claim minimal overhead, the paper does not provide a detailed breakdown of the costs associated with embedding computation, vector search, and forest maintenance relative to performance gains.
2. Limited Evaluation Under High Concurrency: The main benefit of cross-session KV cache sharing appears in high-throughput environments, yet the paper lacks experiments simulating batch inference or parallel request scenarios.
3. Hyperparameter Sensitivity Not Addressed: The system likely depends on hyperparameters like τ_sim, α, and β, but the paper does not include an ablation study or sensitivity analysis to show robustness.

---

> ### Author Rebuttal · Authors · 2025-07-31
>
> Thank you for acknowledging **the significance of the crucial problem** we address, as well as the **novel, holistic and thoughtful system-algorithm co-design**. Below, we provide point-by-point responses to your constructive and helpful questions.
>
> **Q1: Overhead Analysis of SmartCache**
>
> **Response:** Thank you for the comment. As shown in Table 1, the computational and memory overhead with Mistral-7B-Instruct-v0.2 on CoQA. The computational overhead for processing a cache-missed new query (i.e. setting up a new semantic node + performing LLM inference) is approximately 8.8% (18.54ms out of 210.37ms). Due to SmartCache's hierarchical semantic information and co-design with KV cache management, there is a higher likelyhood of reusing cached responses resulting in more substantial performance gains. SmartCache introduces about 7.1% GPU memory overhead (embedding model compared to LLM weights and compute buffer) and little CPU memory overhead (28.05MB).
>
> Table 3: Overhead Breakdown
>
> | Breakdown                    | Computational Overhead | Memory Overhead           |
> |------------------------------|------------------------|---------------------------|
> | Embedding Generation         | 17.13ms                | 1.06GB GPU memory         |
> | Vector Search                | 0.11ms                 | 18.16MB CPU memory        |
> | Vector Insertion             | 1.01ms                 | Shared with vector search |
> | Semantic Forest Maintenance | 0.29ms                 | 9.89MB CPU memory         |
>
>
> **Q2: Performance under High Concurrency**
>
> **Response:** Thank you for the insightful comment. Table 2 shows the concurrency overhead of SmartCache in the scenario with varying number of concurrent queries, based on Mistral-7B-Instruct-v0.2 and CoQA.
>
> Table 2: Overhead analysis under different number of concurrent requests
>
> | # Concurrent Request        | 1     | 2     | 4     | 8     | 16    | 32    | 64    |
> |-----------------------------|-------|-------|:------|:------|:------|:------|:------|
> | Embedding Generation        | 17.13 | 18.00 | 19.44 | 19.54 | 20.81 | 22.08 | 23.32 |
> | Vector Search               | 0.11  | 0.23  | 0.26  | 0.36  | 0.75  | 5.64  | 9.35  |
> | Vector Insersion            | 1.01  | 1.01  | 1.02  | 1.05  | 1.06  | 1.22  | 1.62  |
> | Semantic Forest Maintenance | 0.29  | 0.31  | 0.31  | 0.33  | 0.67  | 1.34  | 2.97  |
>
> The primary source of overhead, embedding generation, supports efficient batching. 8 CPU threads are used for faiss-cpu internal threading and semantic forest maintenance. Contention mainly occurs when multiple queries attempt to add new child nodes to the same parent semantic node, which is mitigated by batching vector insertion.
>
> **Q3: Hyperparameter Sensitivity**
>
> **Response:** Thank you for the comment. The τ_sim is set to 0.75 to give a balanced trade-off between performance and quality. The α and β are selected based on the grid search results. Table 3 presents the ablation results of Qwen2.5-1.5B-Instruct on CoQA, while Table 4 shows the grid search result for α and β with the same configuration as Section 4.4.
>
> Table 3: Different similarity threshold.
>
> | Threshold | EM   | F1   | ROUGE-L | TTFT (ms) | KV Cache (GB) |
> |-----------|------|------|:--------|:----------|:--------------|
> | 0.5       | 17.4 | 35.2 | 35.5    | 26.8      | 4.1           |
> | 0.6       | 21.6 | 41.1 | 41.4    | 27.5      | 4.5           |
> | 0.7       | 23.5 | 45.3 | 45.5    | 30.2      | 6.5           |
> | 0.75      | 24.9 | 46.6 | 47.0    | 31.3      | 7.7           |
> | 0.8       | 25.4 | 47.7 | 47.9    | 49.0      | 8.5           |
> | 0.9       | 25.9 | 48.9 | 49.1    | 83.2      | 12.1          |
>
> Table 4: Different α and β, zipf distribution.
>
> | (α, β)  | (0.1, 0.9) | (0.2, 0.8) | (0.3, 0.7) | (0.4, 0.6) | (0.5, 0.5) | (0.6, 0.4) | (0.7, 0.3) | (0.8, 0.2) | (0.9, 0.1) |
> |----------------------|------------|------------|:-----------|:-----------|:-----------|:-----------|:-----------|:-----------|:-----------|
> | Reuse distance (avg) | 231.8      | 252.0      | 255.9      | 262.3      | 261.6      | 265.4      | 260.5      | 240.9      | 252.5      |
>
> By increasing τ_sim, answer quality can be further improved. However it leads to higher TTFT and increased KV cache usage resulted from reduced opportunities to reuse similar queries.
>
> **Q4: Attention-based Context Switching**
>
> **Response:** Thank you for the comment. Since different models exhibit numeric variations on attention scores, the attention score threshold is profiled for each model offline using five predefined independent queries, averaged across query tokens for each query. The same attention score calculation procedure is triggered during the prefill stage of a qeury when it fails from reusing its child semantic nodes. It is then compared with the the profiled threshold to determine whether or not the query is independent.

---

> > ### Comment · Reviewer_h8B6 · 2025-08-03
> >
> > Thank you for the detailed rebuttal. Your response addresses most of my concerns. But for high concurrency, beyond the overhead analysis, I think smartcache should have greater speedup due to more KV cache reuse opportunities. Could you report how the cache hit rate or TTFT improvement scales with concurrency?

---

> > > ### Author Response · Authors · 2025-08-05
> > >
> > > Thank you for your valuable feedback. We are pleased that our rebuttal addressed most of your concerns. Regarding the performance under high concurrency, we agree that SmartCache should exhibit increased speedup due to enhanced KV cache reuse. Below, we present the TTFT (ms) of SmartCache across varying number of concurrent requests, based on Mistral-7B-Instruct-v0.2 and CoQA.
> > >
> > > | # Concurrent Requests | Baseline | PrefixCache | SmartCache (Speedup over PrefixCache) |
> > > |-------------------------------------------------|----------|-------------|---------------------------------------|
> > > | 1                                               | 190.9    | 179.9       | 36.5 (4.9x)                           |
> > > | 2                                               | 236.1    | 225.2       | 45.7 (4.9x)                           |
> > > | 4                                               | 329.7    | 311.4       | 65.0 (4.8x)                           |
> > > | 8                                               | 501.1    | 487.9       | 96.8 (5.0x)                           |
> > > | 16                                              | 876.2    | 831.8       | 160.4 (5.2x)                          |
> > > | 32                                              | 1631.7   | 1541.9      | 243.3 (6.3x)                          |
> > > | 64                                              | 3084.3   | 2983.5      | 375.3 (7.9x)                          |
> > >
> > > SmartCache delivers increasingly greater speedups under high concurrency. As the number of concurrent requests grows, the LLM’s batching efficiency saturates due to GPU parallelism limits. Nevertheless, SmartCache filters out cache-hit queries, thereby effectively reducing the batch size forwarded to the LLM inference. Combined with KV cache reuse, this significantly cuts down overall computational cost.

---

> > > > ### Comment · Area_Chair_V4fM · 2025-08-05
> > > >
> > > > As the author-reviewer discussion will end on Aug 8, it would be great to leave your further comments for the authors' reply, or leave Mandatory Acknowledgement with your rating.
> > > >
> > > > AC.

---

> > > > ### Comment · Reviewer_h8B6 · 2025-08-05
> > > >
> > > > Thank you for providing the additional experiments. The results are very promising and clearly demonstrate the scalability of SmartCache under high concurrency. Your detailed response has effectively addressed my concerns. Based on this, I have decided to raise my score.

---

> > > > > ### Author Response · Authors · 2025-08-06
> > > > >
> > > > > Thank you very much for your encouraging feedback. We are glad that our additional experiments addressed your concerns. Your support is greatly appreciated.

---

### Note · Authors · 2025-08-13

We sincerely thank all reviewers for their constructive feedback and insightful suggestions.

In our responses, we clarified the design and operation of SmartCache, detailed its computational and memory overhead, and provided additional experiments with larger models and varying concurrency levels. We analyzed the trade-offs between performance, quality, and KV cache usage, explored hyperparameter settings, and addressed scalability and privacy handling concerns.

We also presented detailed concurrency experiments, showing SmartCache's significant speedup under high concurrency level. Additionally, we provided an in-depth overhead breakdown, highlighting the sources of computational and memory cost, as well as contention points and batching strategies to alleviate them.

In the camera-ready version, we will include these additional results on scalability, overhead breakdown and hyperparameter settings. We will add detailed analysis of attention-threshold across more LLM architectures, and the related works discussion. These additions will further strengthen the empirical depth and practical applicability of our work.

---

### Decision · Program_Chairs · 2025-09-17

**Decision:**

Accept (poster)

**Comment:**

This paper proposes Smart Cache, a system for improving the efficiency of multi-turn conversations by reusing cached contexts. The reviewers found that the authors addressed most concerns during the rebuttal, and the overall sentiment was positive, with all reviewers landing at borderline accept or above. The approach is viewed as promising, and the paper makes a clear contribution toward improving efficiency in multi-turn dialogue systems.

However, several points remain only partially resolved. The evaluation was conducted at a limited scale, and the qualtiy degradation and trade-offs in efficiency gains are not fully convincing. In addition, privacy implications were also raised, but the responses were somewhat insufficient. In particular, the AC believes that demonstrating effectiveness in actual multi-turn scenarios would be important. Given the mix of strengths and weaknesses, the paper will be forwarded to the SAC discussion. If accepted, it would be important for the authors to integrate the rebuttal clarifications into the main paper to strengthen its contribution.